# Fault Diagnosis and Fault-Tolerant Control of Permanent Magnet Synchronous Motor Position Sensors Based on the Cubature Kalman Filter

**DOI:** 10.3390/s25196030

**Published:** 2025-10-01

**Authors:** Jukui Chen, Bo Wang, Shixiao Li, Yi Cheng, Jingbo Chen, Haiying Dong

**Affiliations:** 1School of New Energy and Power Engineering, Lanzhou Jiaotong University, Lanzhou 730070, China; 12232215@stu.lzjtu.edu.cn; 2Lanzhou Wanli Airlines Electromechanical Limited Liability Company, Lanzhou 730070, China; wangb087@avic.com (B.W.); chenjb002@avic.com (J.C.); 3School of Automation, Northwestern Polytechnical University, Xi’an 710129, China

**Keywords:** electromechanical actuator, PMSM, position sensor fault, CKF state reconstruction, fault-tolerant control

## Abstract

To address the issue of output anomalies that frequently occur in position sensors of permanent magnet synchronous motors within electromechanical actuation systems operating in harsh environments and can lead to degradation in system performance or operational interruptions, this paper proposes an integrated method for fault diagnosis and fault-tolerant control based on the Cubature Kalman Filter (CKF). This approach effectively combines state reconstruction, fault diagnosis, and fault-tolerant control functions. It employs a CKF observer that utilizes innovation and residual sequences to achieve high-precision reconstruction of rotor position and speed, with convergence assured through Lyapunov stability analysis. Furthermore, a diagnostic mechanism that employs dual-parameter thresholds for position residuals and abnormal duration is introduced, facilitating accurate identification of various fault modes, including signal disconnection, stalling, drift, intermittent disconnection, and their coupled complex faults, while autonomously triggering fault-tolerant strategies. Simulation results indicate that the proposed method maintains excellent accuracy in state reconstruction and fault tolerance under disturbances such as parameter perturbations, sudden load changes, and noise interference, significantly enhancing the system’s operational reliability and robustness in challenging conditions.

## 1. Introduction

Permanent magnet synchronous motors (PMSMs) are widely used in high-reliability applications such as aerospace and precision manufacturing due to their significant advantages, including high efficiency and high-power density [1,2,3,4]. However, under harsh conditions—such as extreme temperatures and strong vibrations—the likelihood of faults occurring in various motor system components increases, with position sensor faults being among the most common [5,6,7]. As a critical element of the motor control system, the position signal directly influences key control algorithms, including coordinate transformations. Sensor faults can cause a severe decline in system performance or even complete failure, posing a serious threat to high-reliability applications.

Current position sensor fault diagnosis methods are primarily categorized into two technical approaches: hardware redundancy and analytical redundancy [8]. Hardware redundancy enhances system reliability by incorporating backup sensors; however, it increases system size, cost, and complexity, making it challenging to meet the requirements for lightweight design and high integration. Conversely, analytical redundancy methods generate redundant information by establishing mathematical models of the system without the need for additional hardware, gradually becoming the mainstream in research. These methods include signal-based, model-based, and knowledge-based approaches [9], with model-based methods offering a slight advantage in designing fault-tolerant control strategies. Nevertheless, existing methods still require improvements in adaptability to dynamic operating conditions, multi-fault handling capabilities, and engineering implementation complexity, underscoring the urgent need to develop more advanced and flexible fault diagnosis and fault-tolerant control technologies.

In recent years, numerous scholars have conducted in-depth research on fault diagnosis and fault-tolerant control of PMSM position sensors. Reference [10] proposes a diagnosis and fault-tolerant control method by analyzing the impact of current and position sensor faults on system performance; however, its diagnostic accuracy is relatively low in low-speed regions. Reference [11] achieves fault identification by utilizing the relationship between sensor detection signals and induced current; nevertheless, it relies on a dual-sensor configuration, resulting in increased system complexity and limited stability. Reference [12] indirectly diagnoses position faults based on rotor internal resistance and estimated current but does not propose a corresponding fault-tolerant control scheme. Reference [13] presents an integrated method of fault detection, reconstruction, and fault-tolerant control based on sliding mode control, enabling real-time fault handling; however, its robustness and computational efficiency in multi-fault coupling scenarios require improvement. Reference [14] combines generalized likelihood ratio testing with structural analysis to achieve highly sensitive early detection of inter-turn short circuits and sensor faults, but it strongly depends on model accuracy, and its reliability under strong noise conditions requires further verification. Reference [15] proposes a static fault diagnosis method based on current estimation, identifying offset faults through current residuals; it demonstrates good robustness at low speeds but exhibits weak adaptability to parameter variations and load disturbances. Reference [16] employs a high-frequency injection method for zero and low-speed conditions, estimating rotor position through high-frequency response and maintaining 90% of rated torque at 5 rpm; however, it introduces additional copper losses and electromagnetic interference and demands a high inverter switching frequency.

To enhance estimation accuracy and control performance, various advanced observers have been developed. Reference [17] employs an Active Disturbance Rejection Controller (ADRC) to replace the traditional sliding mode observer. It estimates the rotor position using an extended back electromotive force (EMF) model and extracts speed and position signals through a phase-locked loop (PLL). This approach effectively mitigates the chattering and lag issues associated with sliding mode observers, thereby improving steady-state performance; however, its robustness against variations in permanent magnet parameters remains to be verified. Reference [18] proposes a sensorless fault-tolerant control method based on a back-EMF observer, enabling continuous system operation following sensor failure. Experimental results demonstrate its effectiveness in steady-state conditions but reveal significant position estimation errors at low speeds or during transient processes. Reference [19] designs a robust state observer to achieve real-time detection and isolation of position and current sensor faults, ensuring stable system operation after faults occur. Nevertheless, under high dynamic loads, the observer’s convergence speed is relatively slow, which adversely affects transient response.

In the field of signal processing and model reconstruction, reference [20] proposes a sensorless fault-tolerant control scheme for PMSMs based on the separation of positive and negative sequence components. It employs zero-crossing detection and rotating coordinate system component extraction to achieve fault monitoring and position reconstruction. Experimental results validate its effectiveness; however, the real-time performance of the algorithm under high-speed or sudden load changes was not evaluated. Reference [21] develops fault feature quantities based on the input/output power relationship to enable online detection of sensor deflection faults and estimation of error angles. This method is suitable for the vibration environment of electric vehicles but relies on power calculations, which may reduce detection sensitivity under low-load conditions. Reference [22] presents a sensorless fault-tolerant strategy based on current space vector error reconstruction. When encoder faults occur, the rotor position is estimated through current error, enabling smooth switching. However, this approach is sensitive to current sampling accuracy and system delay, and its performance degrades under low signal-to-noise ratio conditions.

To further optimize the fault-tolerant process, reference [23] employs a vector tracking observer to achieve fault tolerance for Hall sensors. During faults, the system switches to an open-loop observer to suppress current surges, supporting dual Hall sensor fault handling and significantly improving transient performance. However, open-loop operation may lead to short-term accumulation of estimation errors. Reference [24] proposes a rapid fault diagnosis method based on a pseudo-acceleration change threshold, combined with an improved fault-tolerant interpolation method and an adaptive notch angle observer. This approach effectively addresses diagnostic delays and control accuracy degradation caused by installation deviations in permanent magnet synchronous motor (PMSM) Hall position sensor faults but does not consider complete sensor failure scenarios without position feedback control strategies. Reference [25] detects faults by analyzing the relationship between Hall signal transition edges and system states and reconstructs position information through harmonic analysis. It supports fault tolerance for single or dual Hall sensor failures, maintaining good system performance; however, the computational burden is relatively high, limiting its application in high-frequency scenarios. Reference [26] addresses open-circuit faults in five-phase PMSMs by combining sliding mode control with a nonlinear extended state observer (NESO), significantly suppressing torque fluctuations, but does not cover other fault types such as inter-phase short circuits.

Building upon previous research, this paper addresses the vulnerability of position sensors in permanent magnet synchronous motors (PMSMs) within electromechanical actuation systems to failures under complex operating conditions. It proposes an integrated fault diagnosis and fault-tolerant control method based on the Cubature Kalman Filter (CKF). The objective is to enable precise identification of multiple typical and compound faults while ensuring stable system operation under fault conditions. The main research questions addressed in this paper include: (1) How to establish a unified mathematical model encompassing both basic and compound fault types; (2) how to achieve high-precision and robust reconstruction of motor states; (3) how to design a diagnostic mechanism capable of distinguishing among multiple fault modes; and (4) how to ensure the effectiveness of the fault-tolerant control strategy under various disturbance conditions. The key innovations of this work are as follows: (1) A comprehensive fault model is constructed, covering signal disconnection, signal stalling, signal offset, intermittent disconnection, and coupled disconnection-offset faults—capturing the full spectrum of anomalies commonly encountered in practical engineering scenarios. (2) A CKF-based state reconstruction method utilizing innovation and residual sequences is proposed, enabling highly accurate and robust estimation of rotor position and speed. (3) A dual-parameter threshold-based fault diagnosis mechanism is designed, which not only effectively identifies and differentiates multiple fault patterns but also synchronously triggers a robust fault-tolerant control strategy. The reliability and adaptability of the proposed method are validated through simulations under multiple disturbance scenarios. The remainder of this paper is organized as follows: Section 2 summarizes typical fault types of position sensors, establishes a unified mathematical model, and analyzes the impact of these faults on key motor state variables. Section 3 presents the CKF-based state reconstruction approach, accompanied by Lyapunov stability analysis and computational complexity evaluation. Section 4 details the design of the fault diagnosis mechanism and fault-tolerant control strategy, followed by simulation verification and robustness analysis. Section 5 concludes the paper with a summary and outlook.

## 2. PMSM and Position Sensor Modeling

### 2.1. PMSM Modeling

The mathematical model of the permanent magnet synchronous motor (PMSM) comprises the following equations: the voltage equation, the flux linkage equation, the torque equation, and the motion equation [27]. For analytical purposes, the PMSM is assumed to be an ideal motor that satisfies the following conditions: (1) neglecting saturation of the motor iron core; (2) ignoring eddy current and hysteresis losses in the motor; and (3) assuming the motor current is a balanced three-phase sinusoidal current.

The mathematical model of the surface-mounted permanent magnet synchronous motor is developed in the stationary reference frame.

Voltage equation:(1)$$\left\{\begin{array}{l} u_{\alpha }=Ri_{\alpha }+L_{s}\frac{di_{\alpha }}{dt}-\omega_{e}\psi_{f}\sin\theta \\ u_{\beta }=Ri_{\beta }+L_{s}\frac{di_{\beta }}{dt}+\omega_{e}\psi_{f}\cos\theta \end{array}\right.$$

Electromagnetic torque equation:(2)$$T_{e}=\frac{3}{2}p_{n}\left(\psi_{\alpha }i_{\beta }-\psi_{\beta }i_{\alpha }\right)$$

The equation for the stator flux linkage is given by:(3)$$\left\{\begin{array}{l} \frac{d}{dt}\psi_{a}=u_{a}-Ri_{a}\\ \frac{d}{dt}\psi_{\beta }=u_{\beta }-Ri_{\beta } \end{array}\right.$$

Among them, $\psi_{\alpha }$ and $\psi_{\beta }$ are the flux linkage equations in the stationary coordinate system, respectively.

Equation of Mechanical Motion:(4)$$J\frac{d}{dt}\omega_{m}=\left(\psi_{a}i_{\beta }-\psi_{\beta }i_{\alpha }-T_{L}\right)$$(5)$$\frac{d}{dt}\theta_{m}=\omega_{m}$$

### 2.2. Position Sensor Fault Modeling

As a key component of motor systems, position sensors often operate for extended periods in harsh environments or under severe conditions, which significantly increases the likelihood of failure. Common faults of position sensors and their causes are presented in Table 1.

Due to the diverse types of position sensors, the principles for obtaining position information and the calculation methods vary, resulting in a lack of universality in fault modeling approaches based on working principles. Based on the fault phenomena listed in Table 1, this paper categorizes fault types into typical single faults and complex composite faults and establishes corresponding unified mathematical models.

#### 2.2.1. Mathematical Model of Single Faults

(a) Mathematical Model of Signal Disconnection Fault(6)$$y=\left\{\begin{array}{l} m,\;0\leq t\leq t_{0}\\ 0,\;\;t\geq t_{0} \end{array}\right.$$

(b) Mathematical Model of Signal Stagnation Fault(7)$$y=\left\{\begin{array}{c} m,\;\;0\leq t\leq t_{0}\;\\ a,\;\;\;t_{0}\;\leq t\leq t_{1}\\ m,\;\;\;\;\;\;\;t\geq t_{1} \end{array}\right.$$

(c) Mathematical Model of Signal offset Fault(8)$$y=\left\{\begin{array}{l} m,\;\;\;\;\;\;\;0\leq t\leq t_{0}\;\\ m+C,\;\;\;t_{0}\;\leq t \end{array}\right.$$
where *t*_0_ is the fault occurrence time, *t*_1_ is the fault clearance time; *y* is the output signal value of the position sensor at time *t*; *a* is the position signal value during a signal stagnation fault; and *C* is the position signal offset during a signal offset fault.

#### 2.2.2. Mathematical Model of Complex Faults

(a) Intermittent Disconnection and Recovery Cycling Fault

This model describes a cycling fault caused by intermittent poor contact.(9)$$y(t)=\left\{\begin{array}{l} m(t),\;\;\;if(t-t_{0})\mathrm{mod}T>T\cdot D\;\\ 0,\;\;\;\;\;\;else \end{array}\right.$$
where *t*_0_ is the fault onset time, *T* is the fault period, and *D* is the duty cycle representing the fault duration.

(b) Signal Disconnection—Offset Coupled Fault

This model describes a fault scenario in which a signal disconnection occurs initially, followed by a fixed offset upon signal restoration.(10)$$y(t)=\left\{\begin{array}{l} m(t),\;\;\;\;\;\;\;\;\;0\leq t\leq t_{0}\;\\ 0,\;\;\;\;\;\;\;\;\;\;\;\;t_{0}\;\leq t\leq t_{1}\\ m(t)+C,\;\;\;\;\;\;t\geq t_{1} \end{array}\right.$$
where *t*_0_ is the time at which the disconnection fault occurs, *t*_1_ is the time at which the coupled fault initiates, and *C* is the magnitude of the positional offset in the signal caused by the offset fault.

### 2.3. Impact of Position Sensor Failure on the System

The accuracy of rotor position information is essential for achieving decoupled control in Field-Oriented Control (FOC). As illustrated in Figure 1, rotor angle errors (Δ*θ*) caused by position sensor faults directly compromise the accuracy of the Park and inverse Park transformations, resulting in degraded system performance.

#### 2.3.1. d-q Axis Decoupling Failure

When there is an error Δ*θ* between the angle $\hat{\theta }e$ used by the controller and the actual angle *θ*_e_, the relationship between the actual d-q axis currents and the currents output by the controller is as follows:(11)$$\left\{\begin{array}{l} i_{d}={\hat{i}}_{d}\cos\Delta \theta -{\hat{i}}_{q}\sin\Delta \theta \\ i_{q}={\hat{i}}_{d}\sin\Delta \theta +{\hat{i}}_{q}\cos\Delta \theta \end{array}\right.$$

This coupling relationship causes id and iq to become interdependent, resulting in mutual interference between the magnetic field and torque components, which ultimately leads to the failure of decoupled control.

#### 2.3.2. Electromagnetic Torque Distortion

When the i_d_ = 0 control strategy is employed and an angle error is present, the actual electromagnetic torque is given by:(12)$$T_{e}=\frac{3}{2}p\psi_{f}i_{q}=\frac{3}{2}p\psi_{f}(i_{q}\cos\Delta \theta )$$

The torque gain decreases, and harmonic components are introduced, leading to torque ripple as well as increased vibration and noise.

#### 2.3.3. Decline in Dynamic Performance and Stability

Angle error introduces cross-coupling terms into the current control loop, which act as superimposed disturbances and reduce the current tracking bandwidth. This degradation of the inner-loop performance further results in a sluggish dynamic response in the speed control loop and diminished disturbance rejection capability. Moreover, when Δθ exceeds 90°, the torque gain reverses its sign, creating positive feedback that drives the system toward instability.

## 3. Position Sensor State Reconstruction

### 3.1. Design of CKF State Reconstruction

The Cubature Kalman Filter (CKF) is based on the third-order spherical-radial cubature rule and employs a set of cubature points to approximate the mean and covariance of the state in nonlinear systems with additive Gaussian noise, thereby enabling accurate estimation of the system state.

#### 3.1.1. Volume Rule Design

For a four-dimensional system, the spherical-radial cubature rule generates eight cubature points.

#### 3.1.2. Complete CKF Algorithm Process

The CKF algorithm primarily consists of three processes: state prediction, measurement prediction, and state update. The complete algorithm is outlined as follows:

Step 1: State prediction

(a) Covariance Matrix Decomposition.(13)$$S_{k-1|k-1}=\mathrm{chol}(P_{k-1|k-1})$$

(b) The cubature points generated based on the previous time-step estimate and the decomposed covariance matrix are as follows.(14)$$\chi_{i,k-1|k-1}={\hat{x}}_{k-1|k-1}+S_{k-1|k-1}\xi_{i}$$

(c) Propagate the cubature points through the nonlinear model.(15)$$\chi_{i,k|k-1}^{*}=\chi_{i,k-1|k-1}+\frac{T_{s}}{2}(k_{1}^{\left(i\right)}+k_{2}^{\left(i\right)})$$

(d) Compute the predicted state of the system.(16)$${\hat{x}}_{k|k-1}=\frac{1}{8}\sum_{i=1}^{8}\chi_{i,k|k-1}^{*}$$

(e) Compute the covariance matrix of the predicted state.(17)$$P_{k|k-1}=\frac{1}{8}\sum_{i=1}^{8}(\chi_{i,k|k-1}^{*}-{\hat{x}}_{k|k-1}){\left(\chi_{i,k|k-1}^{*}-{\hat{x}}_{k|k-1}\right)}^{T}+Q$$

Step 2: Measurement Prediction

(a) Perform the Cholesky decomposition on the state covariance matrix.(18)$$S_{k-1|k-1}=\mathrm{chol}(P_{k-1|k-1}{,}^{\prime }lower^{\prime })$$

(b) The cubature points generated based on the current time estimate and the decomposed covariance matrix from the previous time step are:(19)$$\chi_{i,k|k-1}={\hat{x}}_{k|k-1}+S_{k|k-1}\xi_{i}$$

(c) Propagate the obtained cubature points through the nonlinear model.(20)$$Z_{i,k|k-1}=H\chi_{i,k|k-1}$$

(d) Compute the system’s predicted measurement.(21)$${\hat{z}}_{k|k-1}=\frac{1}{8}\sum_{i=1}^{8}Z_{i,k|k-1}$$

(e) Compute the covariance matrix of the predicted measurement.(22)$$P_{zz,k|k-1}=\frac{1}{8}\sum_{i=1}^{8}(Z_{i,k|k-1}-{\hat{z}}_{k|k-1}){\left(Z_{i,k|k-1}-{\hat{z}}_{k|k-1}\right)}^{T}+R$$

(f) Compute the cross-covariance matrix between the predicted state and the predicted measurement.(23)$$P_{xz,k|k-1}=\frac{1}{8}\sum_{i=1}^{8}(\chi_{i,k|k-1}-{\hat{x}}_{k|k-1}){\left(Z_{i,k|k-1}-{\hat{z}}_{k|k-1}\right)}^{T}$$

Step 3: State Update

(a) Compute the Cubature Kalman gain using the measurement covariance matrix and the cross-covariance matrix.(24)$$K_{k}=P_{xz,k|k-1}P_{zz,k|k-1}^{-1}$$

(b) The optimal state estimate is:(25)$${\hat{x}}_{k|k}={\hat{x}}_{k|k-1}+K_{k}(z_{k}-{\hat{z}}_{k|k-1})$$

(c) The updated covariance matrix is:(26)$$P_{k|k}=P_{k|k-1}-K_{k}P_{zz,k|k-1}K_{k}^{T}$$

### 3.2. Online Estimation of Noise Covariance Matrices

Considering that fixed noise covariance matrices Q and R can degrade filter performance under varying operating conditions, this paper incorporates an adaptive scheme (ACKF) that adjusts these parameters online based on innovation and residual sequences. By leveraging the statistical properties of these sequences [28], the approach estimates and updates Q and R in real time according to the covariance matching principle.

The innovation sequence $\tilde{y}k$ and its theoretical covariance $C\tilde{y}$ are defined as follows:(27)$${\bar{y}}_{k}=z_{k}-{\hat{z}}_{k|k-1}$$(28)$$C_{\bar{y}}=E[{\bar{y}}_{k}{\bar{y}}_{k}^{T}]=H_{k}P_{k|k-1}H_{k}^{T}+R_{k}$$

The actual covariance estimate of the innovation sequence is computed using a sliding window of length *N*.(29)$${\hat{C}}_{y,k}=\frac{1}{N}\sum_{j=k-N+1}^{k}{\tilde{y}}_{j}{\tilde{y}}_{j}^{T}$$

The value of the sliding window, *N*, is typically chosen within the range of 10 to 50 and can be adjusted according to the system dynamics.

Based on the covariance matching principle, by forcing the real-time estimate to approach the theoretical value, the update law for the measurement noise covariance matrix ***R*** can be derived as follows:(30)$$R_{k}={\hat{C}}_{\bar{y},k}-H_{k}P_{k|k-1}H_{k}^{T}$$

Similarly, define the residual sequence **r**_k_ and compute its covariance estimate $\hat{C}r,k$ as follows:(31)$$r_{k}=z_{k}-{\hat{z}}_{k|k}$$(32)$${\hat{C}}_{r,k}=\frac{1}{N}\sum_{j=k-N+1}^{k}r_{j}r_{j}^{T}$$

The update law for the process noise covariance matrix *Q*_k_ can be computed using the Kalman gain **K**_k_.(33)$$Q_{k}=K_{k}{\hat{C}}_{r,k}K_{k}^{T}$$

To ensure numerical stability and maintain positive definiteness, only the diagonal elements of the covariance matrices are adaptively adjusted in practice, with their values constrained by a lower bound.(34)$$\left\{\begin{array}{l} R_{k}=\max\left(\mathrm{diag}\left({\hat{C}}_{\bar{y},k}-H_{k}P_{zz,k|k-1}H_{k}^{T}\right),R_{\min}\right)\\ Q_{k}=\max\left(\mathrm{diag}\left(K_{k}{\hat{C}}_{r,k}K_{k}^{T}\right),Q_{\min}\right) \end{array}\right.$$
where *R*_min_ and *Q*_min_ are predefined minimum covariance matrices.

By augmenting the standard CKF with the aforementioned adaptive steps, the covariance matrices Q and R is updated in real time and made available for use in the subsequent iteration cycle.

### 3.3. Modeling of a PMSM System Based on the CKF

According to Section 2.1, the nonlinear state equation of the permanent magnet synchronous motor in the stationary reference frame is as follows:(35)$$\left\{\begin{array}{l} \frac{di_{\alpha }}{dt}=-\frac{R}{L}i_{\alpha }+\frac{\omega_{e}\psi_{f}}{L}\sin\theta +\frac{u_{\alpha }}{L}\\ \frac{di_{\beta }}{dt}=-\frac{R}{L}i_{\beta }-\frac{\omega_{e}\psi_{f}}{L}\cos\theta +\frac{u_{\beta }}{L}\\ \frac{d\omega_{e}}{dt}=\frac{1}{J}(T_{e}-T_{L}-B\omega_{e})\\ \frac{d\theta_{e}}{dt}=\omega_{e} \end{array}\right.$$

Selecting the current along the α-β axis as the measurement value for the state equation.(36)$$\left[\begin{array}{c} i_{\alpha }\\ i_{\beta } \end{array}\right]=\left[\begin{array}{cccc} 1 & 0 & 0 & 0\\ 0 & 1 & 0 & 0 \end{array}\right]\left[\begin{array}{c} i_{\alpha }\\ i_{\beta }\\ \omega_{e}\\ \theta_{e} \end{array}\right]+v$$

The state equation of the PMSM system can be expressed as follows:(37)$$\left\{\begin{array}{l} \dot{x}=f(x)+Bu+\omega \\ z=Cx+v \end{array}\right.$$
where *ω* represents the system noise and *v* represents the measurement noise; both are Gaussian white noise processes. Their covariance matrices are *Q* and *R*, respectively.

Among them:$$\begin{array}{c} x=\left[\begin{array}{c} i_{\alpha }\\ i_{\beta }\\ \omega_{e}\\ \theta \end{array}\right],u=\left[\begin{array}{c} u_{\alpha }\\ u_{\beta } \end{array}\right],z=\left[\begin{array}{c} i_{\alpha }\\ i_{\beta } \end{array}\right]\\ f(x)=\left[\begin{array}{c} -\frac{R}{L_{S}}i_{\alpha }+\frac{1}{L_{S}}\omega_{e}\psi_{f}\sin\theta \\ -\frac{R}{L_{S}}i_{\beta }-\frac{1}{L_{S}}\omega_{e}\psi_{f}\cos\theta \\ 0\\ \omega_{e} \end{array}\right],B=\left[\begin{array}{cc} \frac{1}{L_{S}} & 0\\ 0 & \frac{1}{L_{S}}\\ 0 & 0\\ 0 & 0 \end{array}\right],C=\left[\begin{array}{cccc} 1 & 0 & 0 & 0\\ 0 & 1 & 0 & 0 \end{array}\right] \end{array}$$

Using the second-order Runge–Kutta discretization method:(38)$$x_{k+1}=x_{k}+\frac{T_{s}}{2}(k_{1}+k_{2})+w_{k}k_{1}$$
where *T*_s_ is the sampling time.$$\begin{array}{l} k_{1}=A(x_{k})x_{k}+Bu_{k}k_{2}\\ k_{2}=A(x_{k}+T_{s}k_{1})(x_{k}+T_{s}k_{1})+Bu_{k} \end{array}$$

By incorporating the discretized state equation into the CKF algorithm implementation, the position sensor state of the PMSM can be accurately reconstructed.

In practical applications, because the statistical characteristics of system process noise and measurement noise are typically unknown, the covariance matrices of these noises are generally determined based on prior knowledge. In this paper, the initial values are set as follows:$$Q_{0}=\left[\begin{array}{cccc} 0.1 & 0 & 0 & 0\\ 0 & 0.1 & 0 & 0\\ 0 & 0 & 1 & 0\\ 0 & 0 & 0 & 0.1 \end{array}\right],\;R_{0}=\left[\begin{array}{cc} 0.1 & 0\\ 0 & 0.1 \end{array}\right],P_{0}=\left[\begin{array}{cccc} 0.01 & 0 & 0 & 0\\ 0 & 0.01 & 0 & 0\\ 0 & 0 & 0 & 0\\ 0 & 0 & 0 & 0.01 \end{array}\right]$$

### 3.4. Strict Lyapunov Stability Analysis

#### 3.4.1. Dynamic Error System Modeling

Define the reconstruction error as follows:(39)$$e_{k}=x_{k}-{\hat{x}}_{k|k}$$

The error dynamic equation is:(40)$$e_{k+1}=(\Phi_{k}-K_{k}H)e_{k}+\phi_{k}+\psi_{k}$$

Among them, the state transition matrix is:(41)$$\Phi_{k}=I+T_{s}A_{k}+\frac{T_{s}^{2}}{2}A_{k}^{2},\;A_{k}={\left.\frac{\partial f}{\partial x}\right|}_{{\hat{x}}_{k|k}}$$

The higher-order linearization error is:(42)$$\phi_{k}={\left.\frac{T_{s}^{3}}{6}\frac{\partial^{2}f}{\partial x^{2}}\right|}_{x=\xi }(e_{k}⊗e_{k}),\;\xi \in L(x_{k},{\hat{x}}_{k|k})$$

The composite noise is:(43)$$\psi_{k}=w_{k}-K_{k}v_{k}$$

#### 3.4.2. Proof of Lyapunov Stability

**Theorem** **1.**
*Global Exponential Practical Stability: Suppose there exist positive definite matrices **P** and **Q**, and positive constants η, ρ, and γ such that:*

*(a) Matrix inequality*

(44)
$${\left(\Phi_{k}-K_{k}H\right)}^{T}P(\Phi_{k}-K_{k}H)-P\leq -Q-\gamma I$$


*(b) Boundedness of error*

(45)
$$\|\phi_{k}\|\leq \eta \|e_{k}{\|}^{2}$$


*(c) Noise energy limit*

(46)
$$E[\psi_{k}^{T}\psi_{k}]\leq \rho$$


*Then, the estimation error satisfies the following condition:*

(47)
$$E[\|e_{k}{\|}^{2}]\leq \frac{V(0)}{\lambda_{min}(P)}e^{-\gamma k}+\frac{\rho +\eta^{2}}{\lambda_{min}(Q)}$$




**Proofreading Process:**


(a) Construct a Lyapunov function(48)$$V(e_{k})=e_{k}^{T}Pe_{k}$$

(b) Calculate the differential signal(49)$$\Delta V_{k}\leq -e_{k}^{T}(Q+\gamma I)e_{k}+2\|P\|\eta \|e_{k}{\|}^{3}+\|P\|\eta^{2}\|e_{k}{\|}^{4}+\rho$$

(c) When condition $\left\|e_{k}\right\|\leq \frac{\lambda_{\min}(Q)}{3\left\|P\right\|\eta }$ is met:(50)$$\Delta V_{k}\leq -\frac{\gamma }{2}V(e_{k})+\rho$$

By applying the Dynkin formula, stability is demonstrated.

The above stability analysis theoretically guarantees the bounded convergence of estimation errors; however, evaluating the algorithm’s computational and memory complexity is also essential to assess its engineering feasibility.

### 3.5. Algorithm Complexity Analysis

To assess the feasibility of deploying the algorithm on embedded platforms, this paper presents a quantitative analysis of resource utilization and real-time performance for EKF, standard CKF, and Adaptive Cubature Kalman Filter (ACKF), using the TI TMS320F28379D DSP (Texas Instruments, Dallas, TX, USA), a processor widely adopted in industrial control applications with a 200 MHz clock frequency and 512 KB RAM, as the benchmark platform [29].

As shown in Table 2, the ACKF requires approximately 1.35 μs per computational step, consuming only 3.2% of CPU resources (at a 10 kHz sampling rate) and less than 0.5% of memory. While ensuring the highest numerical stability, it still meets stringent real-time requirements. In comparison, although the EKF is faster at 0.9 μs per step, its reliance on analytical derivatives limits its robustness. The standard CKF avoids derivative calculations but lacks adaptive noise estimation capabilities. The ACKF achieves an optimal balance among accuracy, robustness, and real-time performance, making it directly deployable on industrial DSP platforms.

## 4. Position Sensor Fault Diagnosis and Fault-Tolerant Control

Permanent magnet synchronous motors in electromechanical actuator systems typically operate within the medium to high-speed range at rated speeds. Consequently, the likelihood of position sensor failure during zero or low-speed startup is relatively low. This paper primarily focuses on the study of position sensor faults occurring during the medium to high-speed operation phase of permanent magnet synchronous motors used in electromechanical actuators.

### 4.1. Fault Diagnosis and Fault-Tolerant Strategies

This paper introduces a dual-threshold diagnostic mechanism that utilizes position residuals and signal anomaly duration thresholds within the fault diagnosis logic. This approach effectively mitigates misjudgments and missed detections commonly caused by traditional methods relying on a single reference value, which is susceptible to noise and parameter perturbations. As a result, it significantly enhances the robustness of the diagnostic strategy. The diagnostic logic for individual single faults is illustrated in Figure 2 and Table 3. Because complex faults consist of combinations of single faults, separate diagnostic logic for complex faults has not been established.

Using the absolute value of the position signal, |*θ*_e_|, to diagnose disconnection faults in the position signal. When a fault occurs, the position signal rapidly drops and remains at zero. To prevent false alarms, an abnormal time parameter, *T*_threshold_, should be incorporated. During the fault period ΔT, if the absolute value of the position signal |*θ*_e_| equals zero and ΔT is greater than or equal to *T*_threshold_, it can be concluded that the position sensor has a signal disconnection fault, and the fault diagnosis flag is set to a high level.

Using the rate of change in the position signal, d*θ*_e_/dt, to diagnose position signal stalling faults: when a fault occurs, the sensor output signal remains fixed at the value present at the time of the fault and no longer changes. By monitoring whether d*θ*_e_/dt equals zero and incorporating an abnormal time threshold, *T*_threshold_, to prevent false alarms, it can be determined that a position sensor signal stalling fault has occurred if, during the fault duration Δ*T*, the rate of change continuously remains zero and Δ*T* ≥ *T*_threshold_. In this case, the fault diagnosis flag is set to a high level.

Using the position signal residual Δ*θ* to diagnose position signal shift faults. When the system maintains Δ*θ* < *θ*_threshold_, it indicates that the position sensor is functioning normally. A typical characteristic of a signal shift fault is a constant difference between the rotor’s actual position and the position measured by the sensor, with this difference exceeding the threshold *θ*_threshold_. Since the position signal crosses zero once every rotation cycle, an abnormal time parameter threshold, *T*_threshold_, is introduced to prevent false detections. A shift fault can only be confirmed when, during the fault period *t* > *T*_on_, the measured position error exceeds the position residual threshold *θ*_threshold_ and the duration Δ*T* ≥ *T*_threshold_. At this point, the fault diagnosis flag is set to a high level.

As shown in Figure 3, the output of the fault-tolerant control module is governed by the fault flag signal. Its inputs include the position and speed values detected by the sensor, as well as the position and speed values reconstructed through the volumetric Kalman filter. When the position sensor is functioning normally, the fault flag signal is low. In this state, the fault-tolerant control module uses the sensor-detected values for the motor’s vector control. However, when the fault detection module sets the fault flag signal high, the fault-tolerant control module switches to fault-tolerant mode, using the CKF-reconstructed values for vector control to ensure stable motor operation.

### 4.2. Simulation Verification

A simulation model was developed in the MATLAB/Simulink (2023b) environment to validate the proposed fault diagnosis and fault-tolerant control strategy. Within the standard Field-Oriented Control (FOC) framework, the outer speed loop controller utilizes three alternatives: conventional PI control, Sliding Mode Control (SMC), and an Improved Sliding Mode Control (ISMC). The inner current loop employs a PI controller. The simulation uses a fixed-step oed3 solver with a sampling frequency of 10 kHz and a total simulation duration of 0.5 s. The position signal fault detection threshold is set to 10°, and the abnormal duration threshold is 0.2 s. Table 4 lists the relevant parameters of the PMSM, while Table 5 presents the controller parameters.

#### 4.2.1. Performance Analysis of State Reconstruction Under Normal Operating Conditions

When the position sensor operates normally, the system simultaneously runs the Adaptive Cubature Kalman Filter (ACKF) with adaptive parameters, the standard Cubature Kalman Filter (CKF) with fixed parameters, and the Extended Kalman Filter (EKF) with fixed parameters to reconstruct the rotor position state. This approach aims to pre-evaluate the performance of each algorithm and select the optimal backup algorithm for seamless switching in the event of a fault.

Figure 4 illustrates the reconstructed trajectories and error characteristics of the three algorithms. Figure 4a displays the time-domain response curves of the measured rotor position values alongside the reconstructed values from each filter, while Figure 4c presents the corresponding reconstruction results for rotor speed. As shown in Figure 4b,d, the position reconstruction errors for all three filtering algorithms remain within ±0.2 rad, and the speed reconstruction errors are within 1 r/min. Notably, the ACKF demonstrates the smallest error magnitude and the least fluctuation, indicating superior tracking stability. Furthermore, as depicted in Figure 4e, the introduction of a dual-criterion mechanism combining a time delay threshold with a position residual threshold enables the system to effectively suppress false alarms caused by filter tracking delays or transient disturbances during normal operation.

For a more comprehensive quantitative comparison, Figure 5 systematically evaluates performance across three dimensions: state estimation accuracy, computational efficiency, and convergence speed. The corresponding data are summarized in Table 6. As illustrated in both the figure and the table, the ACKF outperforms other state reconstruction algorithms by achieving the highest estimation accuracy, greatest computational efficiency, and fastest convergence speed, thereby demonstrating significant advantages in all evaluated aspects.

Based on the above analysis, this paper selects the ACKF algorithm as the backup reconstruction method for position sensor faults, ensuring seamless and high-precision state recovery in the following five fault scenarios.

#### 4.2.2. Signal Disconnection Fault

Figure 6 systematically illustrates the collaborative response performance of the fault diagnosis mechanism and the fault-tolerant control strategy under a “position signal disconnection fault. As shown in Figure 6a, at 0.25 s, a position sensor signal disconnection fault occurs, causing the sensor output to instantly drop to zero. Upon real-time detection of this abnormal signal by the fault diagnosis module, a signal anomaly duration detection mechanism is immediately activated. When the anomaly duration exceeds the preset threshold, the system sets the fault flag signal to a logic high “1” at 0.27 s, thereby accurately diagnosing the position sensor signal disconnection fault.

As shown in Figure 6b, once the fault flag signal is set at 0.27 s, the position signal seamlessly switches to the reconstructed value derived from the faulty position sensor signal, without any disturbance. At this point, the system transitions to a fault-tolerant operation mode, ensuring continuous and stable operation.

Figure 6c and Table 7 present the dynamic response characteristics under the fault-tolerant operation mode. The diagnostic system achieves an ultra-fast fault detection response time of 20 ms, while state reconstruction ensures continuous control. Although the ISMC strategy experiences the largest speed drop, it achieves non-overshoot and zero steady-state error control within the shortest recovery time of 0.03 s, demonstrating the best overall dynamic performance.

#### 4.2.3. Signal Stagnation Faults

Figure 7 systematically illustrates the coordinated response performance of the fault diagnosis mechanism and fault-tolerant control strategy under the stall fault condition. The fault occurs at 0.1 s, and the system completes diagnosis and achieves smooth switching of the position signal by 0.12 s. Subsequently, the three control strategies, PI, SMC, and ISMC, demonstrate significant differences in dynamic speed recovery, comprehensively validating the effectiveness of the fault diagnosis and fault-tolerant control strategies in scenarios involving sensor signal stagnation.

Figure 7a demonstrates that the diagnostic module accurately detects signal stall characteristics within 20 ms of fault occurrence and promptly outputs the fault flag, thereby satisfying the functional safety requirement for millisecond-level diagnostic response. Figure 7b shows that immediately following diagnostic confirmation, the reconstruction algorithm activates, enabling a smooth, shock-free transition from the measured signal to the estimated trajectory, thus ensuring continuity of the control layer inputs. Figure 7c and Table 8 quantitatively illustrate the following: under PI control, the speed drops by 478 r/min, recovers in 0.052 s, and exhibits an overshoot of 124 r/min, revealing its weak disturbance rejection and integral windup limitations; under SMC control, the speed drops by 538 r/min, recovers in 0.13 s with no overshoot but leaves a steady-state error of 4 r/min, indicating insufficient compensation for model mismatch; under ISMC control, the speed drops by 530 r/min, recovers in 0.13 s, and shows neither overshoot nor steady-state error, highlighting its strong robustness and effective compensation against state stall faults.

#### 4.2.4. Signal Offset Fault

Figure 8 systematically illustrates the coordinated response performance of the fault diagnosis mechanism and fault-tolerant control strategy under the “signal offset fault” condition. The fault occurs at 0.2 s, and the system completes diagnosis and achieves smooth switching of the position signal by 0.22 s. Subsequently, the three control strategies, PI, SMC, and ISMC, exhibit significant differences in dynamic speed recovery, comprehensively validating the effectiveness and superiority of the fault diagnosis and fault-tolerant control strategies in scenarios involving position sensor offset disturbances.

As shown in Figure 8a, the diagnostic module accurately identifies the offset characteristics and outputs a stable flag within 20 ms, thereby meeting the functional safety requirement for millisecond-level response. As illustrated in Figure 8b, the reconstruction algorithm simultaneously achieves a shock-free transition from measured values to estimated values, ensuring continuous control input.

Figure 8c and Table 9 present the speed response data. Under PI control, the speed decreases by 22 r/min, takes 0.05 s to recover, and exhibits an overshoot of 23 r/min, indicating weak disturbance rejection and integral windup issues. Under SMC control, there is no overshoot, and the recovery time is 0.02 s; however, a steady-state error of 6 r/min remains, reflecting insufficient compensation for systemic bias. Under ISMC control, the system also recovers in 0.02 s, with neither overshoot nor steady-state error, demonstrating its precise compensatory capability for offset disturbances through adaptive mechanisms.

#### 4.2.5. Signal Disconnection-Recovery Loop Fault

Figure 9 systematically illustrates the coordinated response performance of the fault diagnosis mechanism and fault-tolerant control strategy under the “signal intermittent disconnection-recovery loop fault” condition. The first fault occurs at 0.15 s, and the system completes diagnosis within 0.02 s; subsequent cyclic faults maintain the same response speed. Upon successful diagnosis, the reconstructed signal seamlessly replaces the sensor input, ensuring continuous control. The speed response differences among the PI, SMC, and ISMC strategies during the initial fault event are significant, comprehensively validating the engineering effectiveness of the fault diagnosis and fault-tolerant strategies in scenarios involving intermittent disconnection-recovery loop faults.

As shown in Figure 9a, the diagnostic module accurately identifies the intermittent signal characteristics and consistently outputs the fault flag within 20 ms, meeting the functional safety requirement for rapid response to repetitive faults. As illustrated in Figure 9b, the position signal switches immediately to the reconstructed trajectory upon fault confirmation, achieving a smooth, shock-free, and lag-free transition, thereby ensuring uninterrupted input to the control layer.

The speed response data are presented in Figure 9c and Table 10. Under PI control, the speed decreases by 405 r/min, takes 0.08 s to recover, and exhibits an overshoot of 121 r/min, indicating weak disturbance rejection capability and integral windup issues. Under SMC control, the speed decreases by 456 r/min, recovers in 0.035 s, and shows no overshoot but leaves a steady-state error of 6 r/min, reflecting steady-state deviation when compensating for high-frequency disturbances. Under ISMC control, the speed decreases by 425 r/min, recovers in 0.03 s, and demonstrates neither overshoot nor steady-state error, highlighting its robust compensation capability for intermittent state loss.

#### 4.2.6. Signal Disconnection-Offset Coupled Fault

Figure 10 systematically illustrates the coordinated response performance of the fault diagnosis mechanism and the fault-tolerant control strategy under the “disconnection-offset coupled fault” condition. The fault evolves in two stages: first, a signal disconnection occurs at 0.1 s, and the system completes the diagnosis by 0.125 s; however, the reconstructed value does not take effect at this point. Second, at 0.25 s, the disconnection is resolved, and a fixed offset is introduced. The system completes the diagnosis by 0.27 s, while the reconstruction algorithm simultaneously compensates for the offset.

As shown in Figure 10a, the diagnostic module rapidly identifies faults and switches modes within 20 ms in both stages, without any confusion or missed alarms. Figure 10b illustrates that the reconstruction algorithm maintains trajectory continuity during the disconnection phase and performs online compensation for system bias during the offset phase, ensuring smooth control input throughout.

The speed response data are presented in Figure 10c and Table 11. Under PI control, during disconnection, the speed drops by 380 rpm, recovers within 0.05 s, and exhibits an overshoot of 123 rpm. During the offset, the speed drops by 132 r/min and recovers within 0.045 s, exhibiting a 95 r/min overshoot, indicating weak disturbance rejection and integral windup. Under SMC control, the recovery time in both stages is 0.03 s with no overshoot; however, steady-state errors of 5 r/min and 6 r/min persist, respectively, indicating limited compensation capability. Under ISMC control, the smallest speed drop during disconnection was 255 r/min, and during the offset, it was 77 r/min. The system recovers in 0.03 s during both stages, with neither overshoot nor steady-state error, highlighting the robust compensation capability of its adaptive mechanism against coupled disturbances. The PI, SMC, and ISMC strategies exhibit significant differences in both stages, comprehensively validating the engineering effectiveness of diagnosis and fault-tolerant control under coupled fault conditions.

### 4.3. Robustness Analysis

To comprehensively evaluate the robustness of the proposed fault diagnosis and fault-tolerant control strategy under non-ideal operating conditions, systematic simulation-based robustness tests were conducted, focusing on three typical scenarios: motor parameter uncertainties, dynamic external load disturbances, and measurement noise interference. Additionally, the impact of diagnostic threshold selection on fault detection sensitivity and false alarm rates was analyzed to verify the stability, reliability, and practical applicability of the algorithm under conditions of parameter drift, strong disturbances, and noisy environments. All tests were performed using the MATLAB/Simulink platform.

#### 4.3.1. Parameter Perturbation

Under prolonged operation or temperature fluctuations, key electrical parameters of permanent magnet synchronous motors (PMSMs) may drift to varying degrees, thereby affecting the accuracy of state reconstruction and the reliability of fault diagnosis. To evaluate the robustness of the ACKF reconstruction algorithm and diagnostic logic against such model uncertainties, two parameter perturbation scenarios, as detailed in Table 12, were designed.

Under both perturbation scenarios, a −30° offset fault in the position sensor is introduced at 0.2 s and persists until the end of the simulation. Figure 11 illustrates the rotor position reconstruction error and the diagnostic flag response curves under these conditions.

Under the aforementioned parameter perturbation scenarios, the position estimation error of the CKF remains stably within ±0.15 rad, and the fault diagnosis module accurately triggers fault-tolerant switching within the preset time threshold, with no missed or false alarms. This confirms the inherent robustness of the ACKF algorithm against model parameter uncertainties, as well as the strong adaptability of the dual-threshold diagnostic mechanism to parameter drift.

#### 4.3.2. Sudden Load Torque Disturbance

At 0.2 s, a sudden 50% increase in the rated load is applied, rising from 3.5 N·m to 5.25 N·m, accompanied by the introduction of a position sensor offset fault of −30°. As shown in Figure 12, under the combined disturbances of the sudden load change and the concurrent fault, the speed decreases by a maximum of approximately 50 r/min. However, with the implementation of fault-tolerant control, stability is restored within 20 ms. The fault diagnosis flag is accurately triggered at 0.22 s, indicating that the diagnostic logic remains unaffected by load disturbances, and the fault-tolerant control maintains strong resistance to load impacts.

#### 4.3.3. Measurement Noise Robustness Test

Zero-mean Gaussian white noise with a signal-to-noise ratio (SNR) of 30 dB was added to the α-β axis current measurement signals to simulate real-world sampling conditions. As shown in Figure 13, although the noise induces slight fluctuations in the position reconstruction error, with a peak-to-peak amplitude of approximately ±0.17 rad, the diagnostic mechanism effectively filters out transient spike interference and prevents false triggering. Consequently, the system maintains stable operation even under noisy conditions.

#### 4.3.4. Threshold Sensitivity Analysis

To verify the robustness of the selected dual thresholds, this paper systematically varies both threshold parameters within a ±20% range: the position residual threshold is adjusted from 8° to 12°, and the anomaly duration threshold is adjusted from 16 ms to 24 ms, with step sizes of 1° and 2 ms, respectively. This results in a total of 25 threshold parameter combinations. For each combination, 60 simulation experiments were conducted, covering five typical fault conditions as well as normal operating states, to statistically evaluate the corresponding Detection Accuracy (DA) and False Alarm Rate (FAR).

As shown in Figure 14, the visual analysis based on the heatmap of diagnostic accuracy (DA) and false alarm rate (FAR) under different threshold combinations indicates that the selected threshold combination (10°, 20 ms) lies within the central region of a “robust plateau” exhibiting excellent performance. With this parameter setting, the system achieves a diagnostic accuracy of 99.2% and a false alarm rate of 1.2%. Further analysis reveals that when both thresholds vary within ±10% of their nominal values, the system maintains high performance, with DA ≥ 98.5% and FAR ≤ 1.5%. These results demonstrate that the selected threshold combination possesses strong robustness and reliability, ensuring efficient and stable system operation under varying working environments and conditions.

## 5. Conclusions and Future Work

### 5.1. Conclusions

This paper addresses common position sensor faults in permanent magnet synchronous motors (PMSMs) used in electromechanical actuation systems by developing a unified mathematical model that encompasses multiple fault modes, including signal disconnection, signal stall, static offset, intermittent interruption, and disconnection-offset coupling. From the perspective of the dynamic propagation mechanisms of state variables, the paper systematically elucidates how each fault mode affects the closed-loop control performance of the motor, thereby establishing a theoretical foundation for fault mechanism analysis and the design of fault-tolerant architectures.

Building on this foundation, an improved adaptive cubature Kalman filter (ACKF) algorithm is proposed to achieve high-precision and robust online reconstruction of rotor position and speed under sensor failure conditions. Theoretical analysis demonstrates that the algorithm possesses global asymptotic stability within the Lyapunov framework. Simulation results further confirm that the proposed algorithm significantly outperforms the conventional extended Kalman filter (EKF) in terms of estimation accuracy, convergence speed, and disturbance rejection capability, providing a highly reliable and real-time state feedback foundation for subsequent fault-tolerant control.

At the level of coordinated implementation of fault diagnosis and fault-tolerant control, this paper innovatively proposes a diagnostic mechanism based on dual dynamic thresholds: “rotor position residual” and “anomaly duration.” This mechanism can accurately identify multiple types of sensor faults within milliseconds while effectively distinguishing transient disturbances from genuine failures, thereby significantly enhancing diagnostic reliability and reducing false alarms.

Furthermore, an integrated fault-tolerant architecture, termed “Reconstruction, Diagnosis, Switching, Stabilization Control,” has been proposed, demonstrating exceptional system robustness and operational continuity under complex conditions such as parameter perturbations, abrupt load changes, and significant measurement noise. On one hand, the parallel-running reconstruction algorithm ensures high-precision, real-time state estimation following a fault; on the other hand, the dual-parameter diagnostic mechanism enables fine-grained fault classification and rapid response, facilitating disturbance-free and smooth switching of control laws. This proposed approach exhibits strong fault tolerance and environmental adaptability, providing an effective solution for fault diagnosis and fault-tolerant control in electromechanical actuation systems operating under complex conditions, with substantial theoretical significance and practical engineering relevance.

### 5.2. Future Work

The current research is primarily based on simulation validation. Although multiple disturbances and noise sources have been introduced to partially emulate real-world operating conditions, physical experiments under harsh environmental conditions, such as extreme temperatures and mechanical vibrations, have not yet been conducted due to laboratory constraints. Consequently, the study cannot fully replicate actual sensor performance degradation or hardware compatibility issues, the practical validity of the conclusions requires further verification through physical experimentation.

The next phase of the project will involve constructing a PMSM experimental platform equipped with real position sensors, such as optical encoders and Hall-effect sensors. Testing will be conducted in an environmental chamber that simulates harsh conditions, including temperature cycling from −55 °C to 125 °C and high-frequency vibration. Measured current waveforms, position signals, and speed responses will be systematically compared with simulation results to comprehensively evaluate the performance of the proposed fault diagnosis and fault-tolerant control methods under realistic, complex scenarios. This approach will further enhance the engineering applicability and practical value of the research.

## Figures and Tables

**Figure 1 sensors-25-06030-f001:**
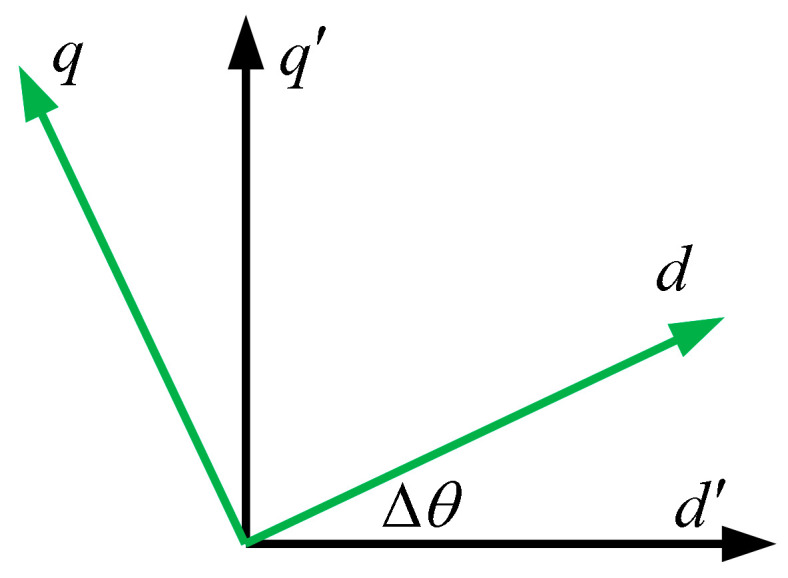
d-q axes during position sensor failure.

**Figure 2 sensors-25-06030-f002:**
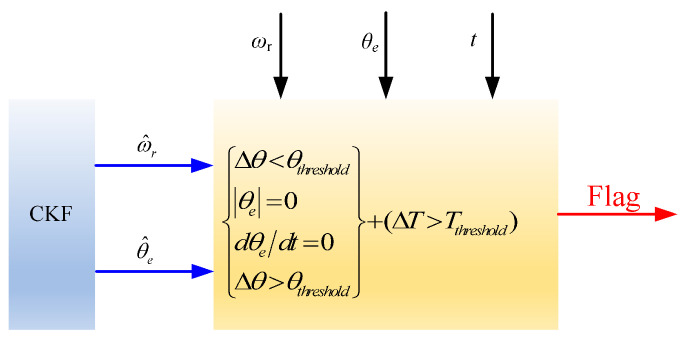
Schematic diagram of position sensor fault diagnosis.

**Figure 3 sensors-25-06030-f003:**
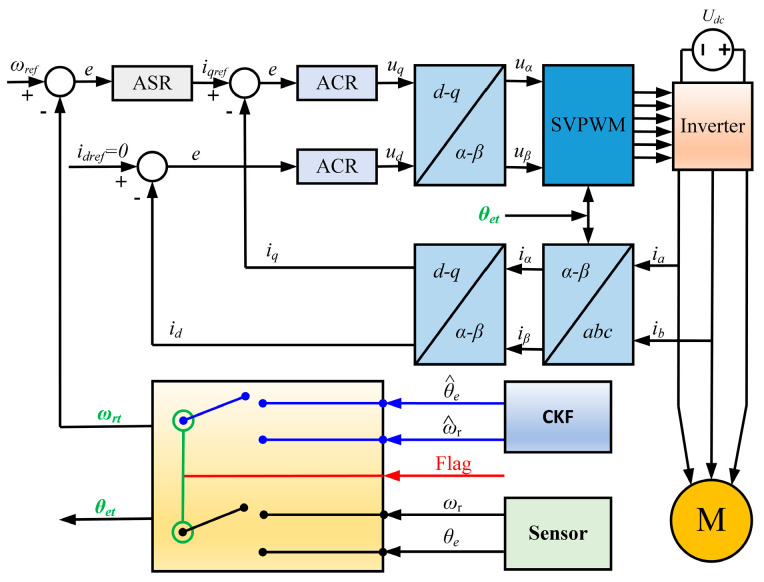
Fault-tolerant vector control schematic diagram.

**Figure 4 sensors-25-06030-f004:**
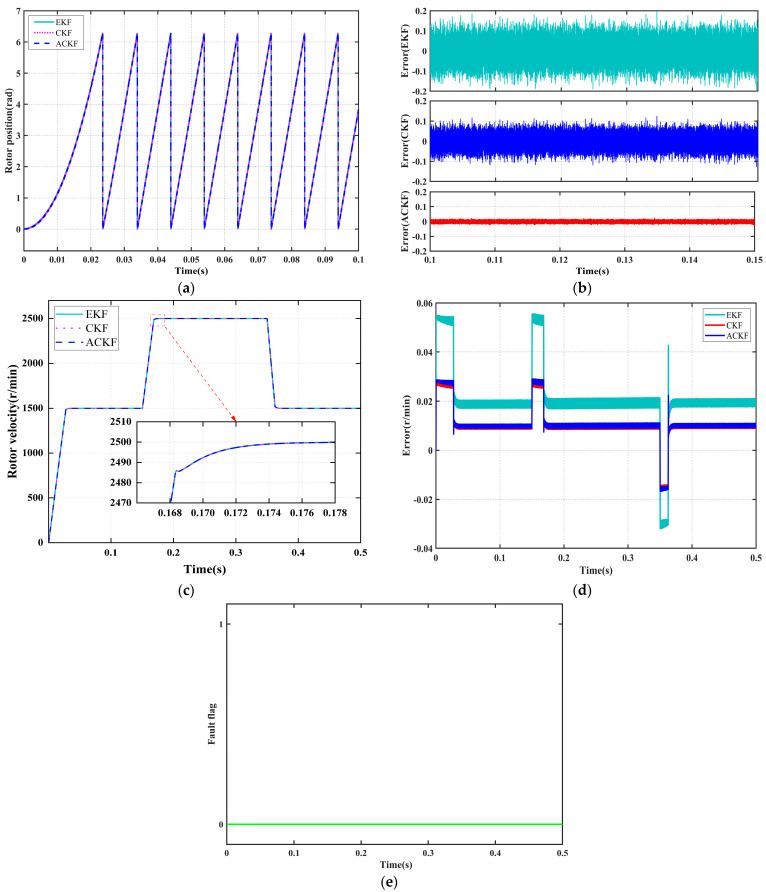
Dynamic response of the system in normal mode. (**a**) Rotor position signal reconstruction. (**b**) Rotor position signal error. (**c**) Rotor velocity reconstruction. (**d**) Rotor velocity error. (**e**) Position Sensor Fault flag.

**Figure 5 sensors-25-06030-f005:**
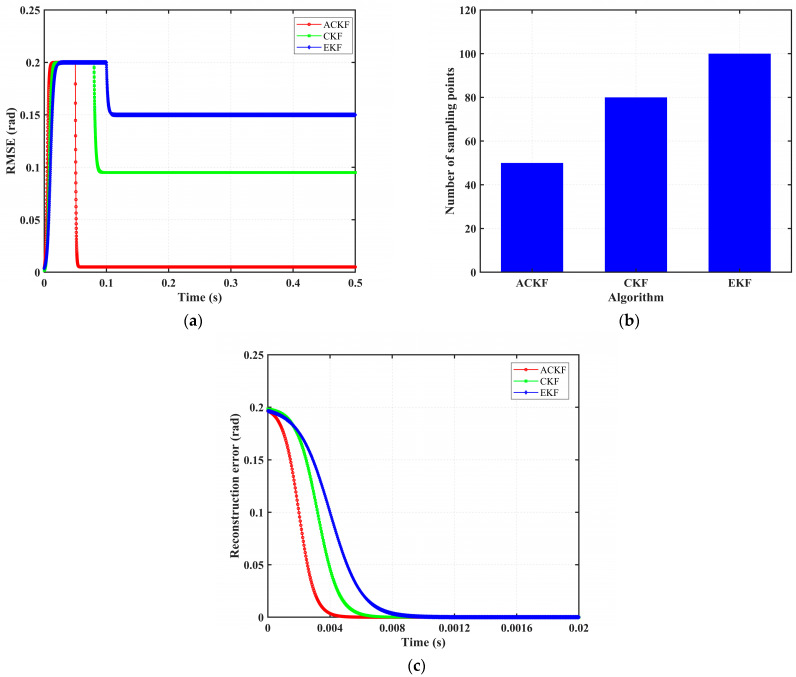
Comparison of state reconstruction algorithm performance under normal operating conditions. (**a**) Reconstruction accuracy. (**b**) Algorithm computational efficiency. (**c**) Algorithm convergence speed.

**Figure 6 sensors-25-06030-f006:**
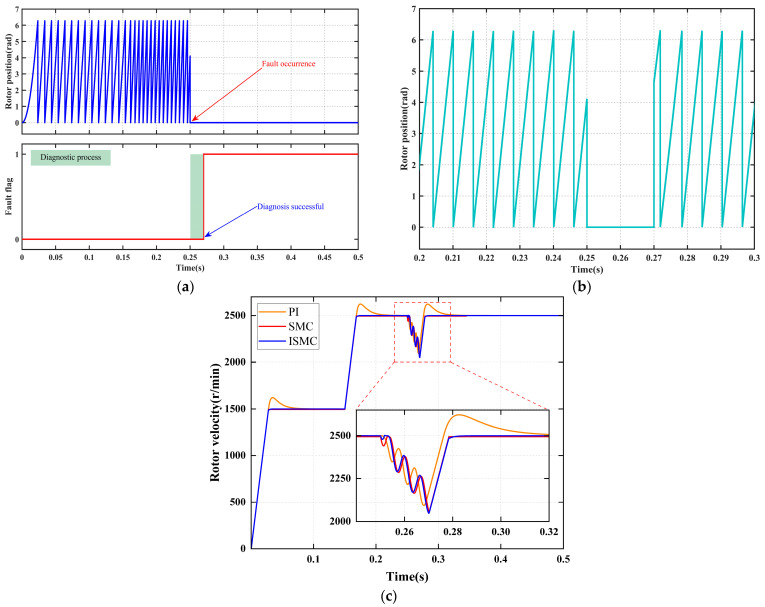
Signal disconnection fault diagnosis and fault-tolerant control. (**a**) Fault diagnosis. (**b**) Rotor position signal. (**c**) Rotor velocity signal.

**Figure 7 sensors-25-06030-f007:**
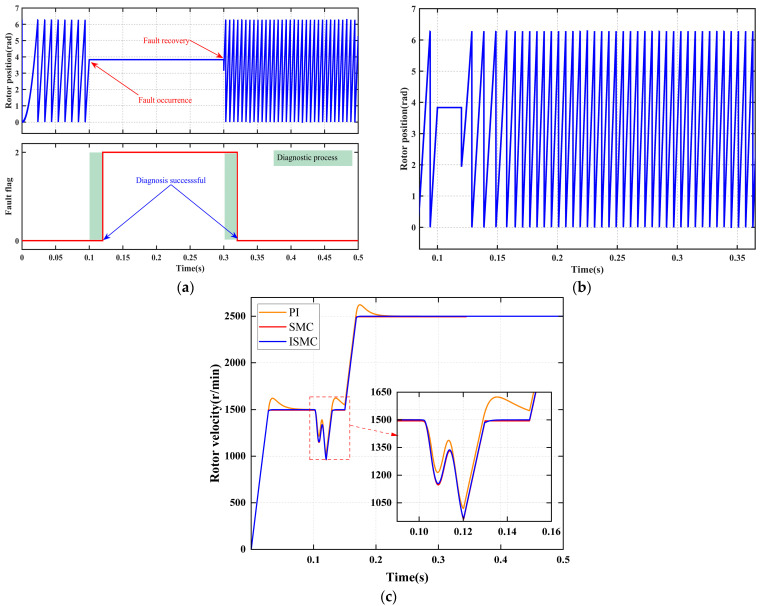
Signal stagnation fault diagnosis and fault-tolerant control. (**a**) Fault diagnosis. (**b**) Rotor position signal. (**c**) Rotor velocity signal.

**Figure 8 sensors-25-06030-f008:**
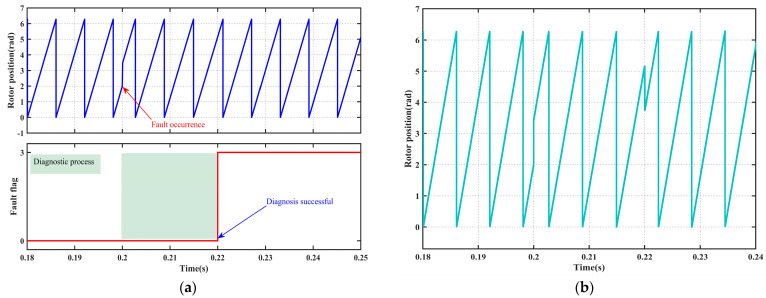
Signal offset fault diagnosis and fault-tolerant control. (**a**) Fault diagnosis. (**b**) Rotor position signal. (**c**) Rotor velocity signal.

**Figure 9 sensors-25-06030-f009:**
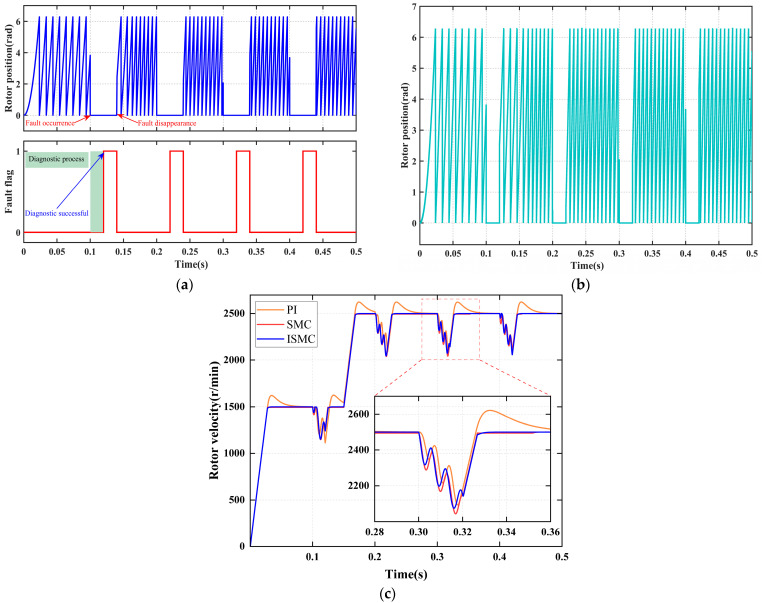
Diagnosis and fault-tolerant control of signal intermittent disconnection-recovery loop Faults. (**a**) Fault diagnosis. (**b**) Rotor position signal. (**c**) Rotor velocity signal.

**Figure 10 sensors-25-06030-f010:**
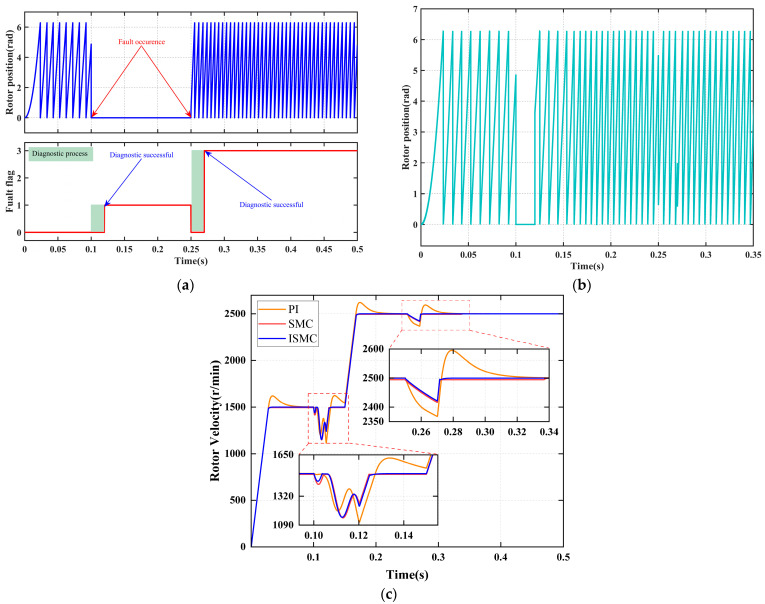
Diagnosis and fault-tolerant control of coupled faults. (**a**) Fault diagnosis. (**b**) Rotor position signal. (**c**) Rotor velocity signal.

**Figure 11 sensors-25-06030-f011:**
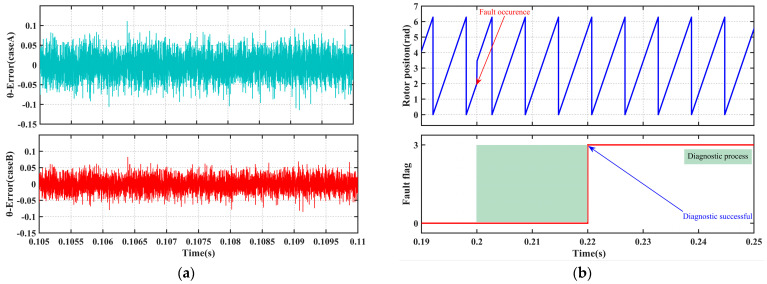
System response under parameter perturbation scenarios. (**a**) State-reconstructed rotor position error. (**b**) Fault diagnosis flag. (**c**) Rotor velocity signal.

**Figure 12 sensors-25-06030-f012:**
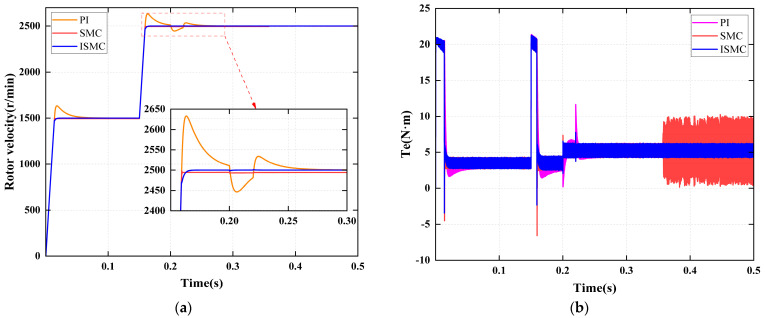
System response to sudden load change. (**a**) Rotor velocity. (**b**) Electromagnetic torque. (**c**) Fault diagnosis flag.

**Figure 13 sensors-25-06030-f013:**
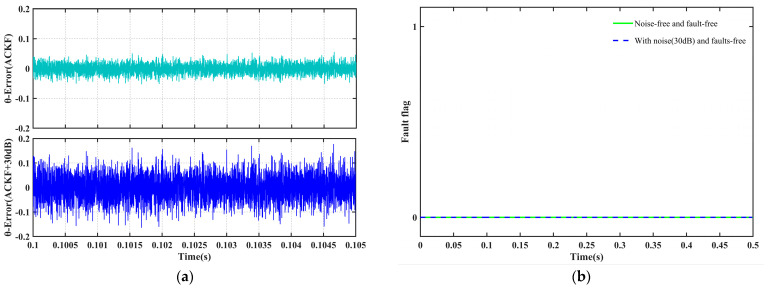
System response under measurement noise conditions. (**a**) Comparison of position reconstruction errors. (**b**) Stability of the fault diagnosis flag.

**Figure 14 sensors-25-06030-f014:**
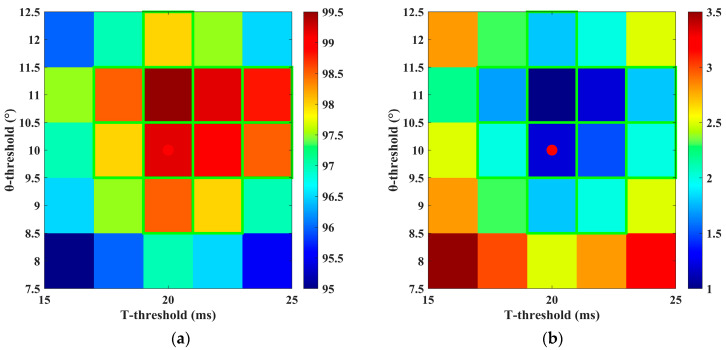
Threshold sensitivity heatmap. (**a**) DA (qualified area green box); (**b**) FAR (qualified area green box).

**Table 1 sensors-25-06030-t001:** Common Fault Phenomena of Position Sensors.

Serial Number	Malfunction Symptoms	Possible Causes of Failure
1	Output signal pulses decrease	Foreign object obstruction
2	Intermittent output signal	Poor contact or loose connection
3	No output signal	Wiring disconnected or device malfunctioning
4	The output signal value remains unchanged	device malfunctioning

**Table 2 sensors-25-06030-t002:** Comparison of Resource Usage and Performance of Nonlinear Filtering Algorithms on the TMS320F28379D Platform (*n* = 4).

Algorithm	FLOPs/Step	Floats	Execution Time per Step (μs)	CPU Utilization	Stability	Requirement Guidance
EKF	≈180	≈64	≈0.9	≈2.1%	Medium	Y
CKF	≈249	≈80	≈1.25	≈2.9%	High	N
ACKF	≈270	≈84	≈1.35	≈3.2%	Highest	N

**Table 3 sensors-25-06030-t003:** Position Sensor Fault Diagnosis.

Operating Status of the Position Sensor	Position Signal Determination	Abnormal Time Determination	Fault Indicator Status Display
Normal Mode	Δ*θ* * < *θ*_threshold_ *	/	0
Signal disconnection fault	|*θ_e_*| = 0	Δ*T* * ≥ *T*_threshold_ *	1
Signal Stagnation Faults	d*θ_e_*/d*t* = 0	Δ*T* ≥ *T*_threshold_	2
Signal offset Fault	Δ*θ* > *θ*_threshold_	Δ*T* ≥ *T*_threshold_	3

* Δ*θ* represents the position error; *θ*_threshold_ denotes the position residual threshold; Δ*T* indicates the fault duration; and *T*_threshold_ is the signal anomaly time threshold.

**Table 4 sensors-25-06030-t004:** Relevant Parameters of the PMSM.

Parameter	Value	Symbol/Unit
Inverter Working Voltage	270	*U*_dc_/V
Stator Inductance	8.5 × 10^−3^	*L*s/mH
Stator Resistance	0.875	*R*s/Ω
Rotor Inertia	1 × 10^−3^	J/kg·m^2^
Pole of Pairs	4	-
Permanent magnet magnetic flux	0.175	*Ψ*_f_/Wb

**Table 5 sensors-25-06030-t005:** Controller Parameters.

PI		Value	SMC	Value	ISMC	Value
Speed loop	Kp	0.25	c	60	ε	0.08
	Ki	18	ε	200	$\eta_{1}$	5000
Current loop	Kp	30	q	0.0005	k_1_	200
	Ki	1000	/	/	k_2_	0.01

**Table 6 sensors-25-06030-t006:** Comparison of Reconstructed Speed and Position.

State Reconstruction Algorithm	Average Positional Error (rad)	Average Speed Error (r/min)
ACKF	0.035	0.01
CKF	0.095	0.015
EKF	0.15	0.02

**Table 7 sensors-25-06030-t007:** Comparison of Dynamic Responses of Different Control Strategies During Fault Conditions.

Control Strategy	Speed Drop (r/min)	Recovery Time (s)	Overshoot (r/min)	Steady-State Error (r/min)
PI	406	0.08	121	0
SMC	438	0.03	0	6
ISMC	452	0.03	0	0

**Table 8 sensors-25-06030-t008:** Comparison of Dynamic Responses of Different Control Strategies During Fault Conditions.

Control Strategy	Speed Drop (r/min)	Recovery Time (s)	Overshoot (r/min)	Steady-State Error (r/min)
PI	478	0.052	124	0
SMC	538	0.03	0	4
ISMC	530	0.03	0	0

**Table 9 sensors-25-06030-t009:** Comparison of Dynamic Responses of Different Control Strategies During Fault Conditions.

Control Strategy	Speed Drop (r/min)	Recovery Time (s)	Overshoot (r/min)	Steady-State Error (r/min)
PI	22	0.05	23	0
SMC	0	0.02	0	6
ISMC	0	0.02	0	0

**Table 10 sensors-25-06030-t010:** Comparison of Dynamic Responses of Different Control Strategies During the First Fault.

Control Strategy	Speed Drop (r/min)	Recovery Time (s)	Overshoot (r/min)	Steady-State Error (r/min)
PI	405	0.08	121	0
SMC	456	0.03	0	6
ISMC	425	0.03	0	0

**Table 11 sensors-25-06030-t011:** Comparison of Dynamic Responses of Different Control Strategies During Fault Conditions.

Fault Time	Control Strategy	Speed Drop (r/min)	Recovery Time (s)	Overshoot (r/min)	Steady-State Error (r/min)
0.1 s	PI	380	0.05	123	0
	SMC	350	0.03	0	5
	ISMC	255	0.03	0	0
0.25 s	PI	132	0.04	95	0
	SMC	83	0.03	0	6
	ISMC	77	0.03	0	0

**Table 12 sensors-25-06030-t012:** Parameter Perturbation Scenarios.

Parameter	Rated Value	Case A	Case B
Rs/Ω	0.875	1.1375/(+30%)	0.65625 (−25%)
Ls/*H*	5.8 × 10^−3^	4.64 × 10^−3^/(−20%)	7.25 × 10^−3^ (+25%)
*ψ_f/_Wb*	0.175	0.14875/(−15%)	0.1925 (10%)

## Data Availability

The original contributions presented in this study are included in the article. Further inquiries can be directed to the corresponding author(s).

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
