# Peer review of "Fault Diagnosis and Fault-Tolerant Control of Permanent Magnet Synchronous Motor Position Sensors Based on the Cubature Kalman Filter"

_sensors, 2025, doi:10.3390/s25196030_

Round 1
Reviewer 1 Report
Comments and Suggestions for Authors
1、Lack of Physical Experiment Verification, and Practicality of Conclusions Needs Enhancement
The paper only verifies the effectiveness of the method through simulations, without conducting physical experiments. The simulation environment cannot fully reproduce complex scenarios in actual harsh working conditions (e.g., high and low temperatures of -55℃~125℃, high-frequency vibrations) such as sensor characteristic attenuation and signal interference, nor can it verify the compatibility between hardware circuits and algorithms. It is recommended to build a PMSM experimental platform, equip it with actual position sensors (e.g., encoders, Hall sensors), test the actual performance of the method in an experimental chamber simulating harsh environments, supplement the comparison between experimental data (e.g., actual current and speed waveforms) and simulation results, and improve the credibility of conclusions.
2、Incomplete Fault Type Coverage, and Complex Fault Scenarios Are Not Considered
The existing research only focuses on three single typical faults ("signal disconnection, stalling, and drift"), and does not involve other common faults of PMSM position sensors (e.g., signal distortion caused by high-frequency electromagnetic interference, "disconnection-recovery" cycle faults caused by intermittent poor contact), let alone multi-fault coupling scenarios such as "disconnection + drift". However, the probability of multi-fault coupling is relatively high in actual engineering. It is recommended to expand fault type modeling, add multi-fault coupling simulations or experiments, and verify the adaptability of the method to complex faults 2-54, .
3、CKF Parameters Not Optimized, and Filtering Performance Has Room for Improvement
The paper uses fixed values for the process noise covariance (Q) and measurement noise covariance (R) of CKF, without explaining the basis for parameter selection, nor considering the impact of working condition changes (e.g., speed increase from 1500 rpm to 2500 rpm) on noise characteristics. Fixed Q and R may lead to reduced filtering accuracy or slow convergence under high dynamic working conditions. It is recommended to supplement the CKF parameter optimization scheme, such as designing an adaptive CKF (adjusting Q and R through real-time residuals) or using intelligent algorithms such as Particle Swarm Optimization (PSO) to optimize initial parameters, and compare the filtering effects of fixed parameters and adaptive parameters to improve the adaptability of the algorithm to working condition changes 2-147, .
4、Lack of Algorithm Comparison and Threshold Sensitivity Analysis
(1)No Comparison with Similar State Estimation Algorithms: The paper emphasizes the advantages of CKF but does not quantitatively compare it with Unscented Kalman Filter (UKF), Extended Kalman Filter (EKF), or Particle Filter (PF) in terms of "state reconstruction accuracy, computational complexity, and convergence speed", making it impossible to clearly highlight the performance advantage boundaries of CKF.
(2)Rationality of Threshold Setting Not Verified: The determination of dual-parameter thresholds (10° position residual, 20 ms abnormal duration) lacks sensitivity analysis, and the impact of "5%~20% fluctuation of thresholds" on diagnosis accuracy and false alarm rate is not discussed, making it impossible to prove the robustness of threshold selection. It is recommended to supplement algorithm comparison experiments and threshold sensitivity analysis to improve the rigor of conclusions 2-194, .
Reviewer 2 Report
Comments and Suggestions for Authors
This paper proposes a fault diagnosis and fault-tolerant control scheme for PMSM position sensors based on the Cubature Kalman Filter, with dual-threshold detection and seamless switching to reconstructed estimates. The topic is relevant and timely, but the paper requires a significant revision to make the research questions explicit, highlight its novelty, strengthen validation, and improve presentation.
-
Make the research questions explicit at the end of the introduction, such as the effectiveness of CKF in reconstructing position signals, the robustness of the dual-threshold logic, and the comparative performance of PI, SMC, and ISMC under faults.
-
Clearly articulate the contribution of the work, emphasizing what is new compared to existing literature, and add quantitative baselines against established methods such as EKF, UKF, or robust observers.
-
Section 3 on field-oriented control is overly didactic. Condense the content, remove basic explanations, and write for a specialist audience.
-
Ensure every equation is numbered and cited in the text—correct instances where equations are presented without reference or numbering.
-
Fix inconsistencies in notation and technical terms, such as “Dyingkin formula” which should be “Dynkin formula”, and standardize symbols like θe, ψf, and α–β/d–q frames.
-
Strengthen validation using robustness analysis against parameter uncertainties, load variations, and measurement noise. Evaluate the sensitivity of the diagnostic logic to threshold values, providing sweeps, detection delays, false positives, and false negatives.
-
Provide technical details to enable reproducibility, including sampling time, Simulink solver, controller gains, and Q/R parameters of the CKF.
-
If possible, add bench or HIL experiments. If not feasible, justify and reinforce comparisons with established methods.
-
Explicitly discuss each subplot in the figures, providing quantitative analysis. Some figures are essential, such as Figure 6, but others appear redundant and should be reconsidered to reduce excessive illustration.
-
Submit figures in vector format (SVG) to preserve quality under zoom and allow textual search.
-
Explain how diagnostic thresholds were defined and discuss their robustness under different operating conditions and noise levels.
-
Include computational complexity analysis of the CKF, such as per-sample cost and feasibility on typical controllers, and compare with EKF and UKF.
-
Improve editorial clarity by reducing dash-heavy sentences and adopting a more natural, technical style.
-
Standardize terminology, fix hyphenation, and remove template leftovers.
-
All authors are recommended to use institutional e-mails and provide an ORCID.
Reviewer 3 Report
Comments and Suggestions for Authors
1, What is responsible for the large rotor transient at start-up in Figure 4?
2, When the fault diagnosis is set to high level, how is the system stability calibrated?
3, How are the proposed closed loop control methods (PI, 499 SMC, ISMC) differentiated from FOC in the study?
4, Authors should expand on the fault-tolerance capability of their proposed control model.
5, How is the system efficiency and cost affected for this control strategy?
Round 2
Reviewer 2 Report
Comments and Suggestions for Authors
I understand that the authors have addressed the requests I made, as well as those from the other reviewers. The topic is relevant, original, and well-structured, it tackles an important gap, and therefore deserves to be published.
Reviewer 3 Report
Comments and Suggestions for Authors
Thanks for the detailed feedback. I am satisfied.